# Effects of Stationary Bikes and Elliptical Machines on Knee Joint Kinematics during Exercise

**DOI:** 10.3390/medicina60030498

**Published:** 2024-03-18

**Authors:** Min-Yan He, Huai-Po Lo, Wei-Han Chen

**Affiliations:** 1Department of Physical Education and Sport Science, National Taiwan Normal University, Taipei 10610, China; 81030017A@ntnu.edu.com; 2Department of Athletics Performance, National Taiwan Normal University, Taipei 10610, China; lbow91@gmail.com; 3Graduate Institute of Sports Equipment Technology, University of Taipei, Taipei 11153, China; 4Department of Physical Education and Kinesiology, National Dong Hwa University, Hualien 974301, China

**Keywords:** inverse dynamics, joint torque, stationary bike, elliptical machine

## Abstract

*Background and Objectives:* This study examined the influence of stationary bikes and elliptical machines on knee movement and joint load during exercise. *Materials and Methods*: Twelve healthy male participants engaged in pedaling exercises on stationary bikes and elliptical machines at speeds of 50 and 70 revolutions per minute (rpm). Knee movement and joint load were assessed using a motion analysis system. *Results*: The results indicated that elliptical machines induced higher knee joint torque compared to stationary bikes. Notably, peak torque occurred at different joint angles, with stationary bikes reaching an earlier peak at 70°–110° and elliptical machines showing a later peak at 135°–180°. Increased pedaling speed correlated with higher peak knee joint torque on both machines. With the elliptical machine, a higher pedaling frequency correlated with increased peak forces on the knee and ankle joints, as well as vertically. Interestingly, both types of equipment were associated with enhanced peak knee joint torques during high-speed pedaling. Conversely, constant pedaling on elliptical machines limited the ankle angle and could induce inward rotation. *Conclusions*: This study focused on knee joint torque variations during pedaling on indoor stationary bicycles and elliptical machines. Elliptical machines showed higher peak values of forces and torque, particularly during the propulsive and recovery phases, indicating potential challenges to the knee joint. Notably, peak pedal angles occurred earlier on indoor stationary bicycles, emphasizing the impact of equipment choice on joint kinetics.

## 1. Introduction

Leisure activities and fitness exercises possess considerable importance among the general public due to their extensive health benefits, entertainment value, accessibility, and inclusivity [1,2]. The incorporation of specialized fitness equipment assists individuals in achieving heightened levels of physical fitness [3]. Therefore, stationary bikes and elliptical machines, among other devices, have been designed to align with the human gait pattern, featuring both the stance phase and swing phase [4]. When using exercise equipment, the motion can be classified into closed kinetic chain exercise (CKCE) and open kinetic chain exercise (OKCE). CKCE involves the distal segment of the limb resisting a significant amount of resistance, while OKCE allows free movement of the distal segment without resistance [5]. Hence, we find that in both indoor cycling bikes and elliptical machines, the predominant mechanism during the pedaling cycle is closed kinetic chain exercise. In contrast to previous studies focusing on the kinematic and kinetic analysis of individual exercise equipment, we conducted a comparative study using two widely adopted exercise machines in the current market: indoor cycling and elliptical trainers. We applied inverse dynamics equations to investigate the differences in knee joint loading between these two exercise machines.

Specifically, cycling tends to subject the knee joints to forces ranging between 0.5 and 1.5 times the individual’s body weight [6,7,8,9,10], while walking or jogging may apply forces amounting to 2.5 times the body weight or, in certain circumstances, even exceeding 6 times the body weight [11,12]. For most users, the critical factors affecting knee loads in cycling are exercise intensity and the cadence itself, but posture on the bike also has an effect [13,14,15,16]. Past studies have indicated that, during cycling, the flexion of the knee joint results in increased force on the patella from the femur, whereas these forces diminish when the knee joint is in extension [17]. This phenomenon is not exclusive to cycling but is associated with the inherent characteristics of the patellofemoral joint. It is noteworthy that, during cycling, the knee is capable of a larger range of motion, reaching up to 70 degrees, thereby causing significant variations in force [18]. An essential aspect to consider is the interplay between the crank rotation speed during cycling and the timing of force peaks on the knee joint. As the crank rotation speed increases from 60 to 105 rpm, the force peak is discerned at a progressively later knee joint angle [19,20], culminating in intensified forces on the knee joint during the later phases of the crank rotation range at elevated speeds. In contrast, lower crank speeds induce the force peak to materialize within the initial segment of the crank rotation range, consequently lessening the forces on the knee joint [21]. Thus, appropriate adjustments to the knee joint angle and crank rotation speed can effectively minimize these forces.

Compared to cycling, the use of an elliptical machine offers a unique advantage in terms of reduced strain on the knee joints, thus rendering it an appealing option for a diverse population, including those with joint sensitivities [22,23]. In elliptical exercise, users typically tend to maintain an average knee joint angle roughly within the range of 30 to 40 degrees. Observations indicate a critical threshold at approximately 40 degrees, beyond which an apparent increase in knee joint pressure may be detected [24]. The complexity increases when considering pedal frequencies during elliptical exercise, as it has been noted to influence the physiological response. More specifically, in fixed-resistance elliptical exercises performed at pedal frequencies of 56, 69, and 80 rpm, all lower limb muscles are actively involved. Notably, a discernible trend is observed where an elevated pedal frequency corresponds with a tendency for the knee joint to be held in a more prominently flexed position [25]. Previous studies indicate that elliptical machine training produces similar kinematic and kinetic patterns compared to walking [26,27] and reduces knee-loading impulses [24]. In comparison to elliptical training, cycling generally results in lower knee joint compressive forces, extension torques, and abduction torques [28].

The bicycle and elliptical are some of the more studied classic machines in this field of research [29,30,31]. Both types of equipment produce diminished stress on the lower extremities [32]. Specifically, the elliptical machine offers distinct advantages by evenly distributing the load across both feet during the pedaling phase, consequently alleviating stress on the lower limb joints [33,34,35]. However, it is imperative to analyze the angle of the lower limb joints and the frequency of pedaling, as these factors can significantly influence the efficacy of these exercise modalities. Extended and repetitive exercise sessions, if excessive, may place undue weight on the lower limb joints, potentially leading to injury and discomfort. Hence, a prudent approach must be adopted to circumvent overuse and the associated adverse effects. Moreover, the existing body of literature, primarily focused on the kinematic analysis of individual exercise equipment, reveals a marked absence of research contrasting the differences between two or more types of equipment. This highlights an unaddressed gap in current understanding.

Given the prevailing concentration on analyzing the kinematics and dynamics of specific exercise devices in contemporary research, there emerges a noticeable scarcity of comparative studies among various types of exercise equipment. Consequently, this study seeks to redress this deficit by examining two widely employed indoor fitness machines: stationary bikes and elliptical machines. Through the application of inverse dynamics equations, this research intends to scrutinize the effects of these machines on knee joint kinematics during exercise, with a particular emphasis on contrasting the knee joint movements induced by both modalities. Future endeavors in this field may explore alterations in traditional pedal structures and trajectories as potential avenues to diminish the risk of lower limb joint injuries. There is a pronounced need for in-depth exploration to evaluate the differences in knee joint torque between stationary bikes and elliptical machines at varying speeds. Such investigative efforts would undoubtedly contribute invaluable insights to sports enthusiasts, sports equipment manufacturers, and engineers alike. The findings could guide future users in the optimal utilization of these exercise machines and inform manufacturers in enhancing the design and functionality of the equipment.

## 2. Materials and Methods

A total of twelve healthy adult males (mean age: 25.08 ± 1.51 years; mean height 174.25 ± 4.54 cm; mean weight: 68.83 ± 6.69 kg), devoid of any history of lower limb neurological, muscular, skeletal, tendon, ligament, or cardiovascular diseases within the preceding six months, were selectively recruited as participants for this study. Prior to their inclusion, they were furnished with comprehensive information pertaining to the research methods, procedures, and precautions. The participants were also provided with detailed instructions to review. After ensuring a complete understanding of the experimental procedures, the participants provided informed consent, thereby formally signifying their voluntary participation.

This article delineates a comparative study focusing on the kinematic parameters of two widely used exercise machines: stationary bikes and elliptical machines. The employed methodological framework consisted of using the Motion Analysis System 3D (SI-660-60, Motion Analysis Corporation, Rohnert Park, CA, USA) to record spatial coordinates and a six-axis force/torque sensor, as depicted in Figure 1 (Six-Axis/Torque Sensor System A3, ATI, Wooster, OH, USA), positioned above the stationary bike in Figure 2 (stationary bicycle/Giant tempo indoor exercise bicycle, Taichung, Taiwan) and elliptical machine in Figure 3 (elliptical machine/Diamondback 460EF, Alamo, CA, USA), utilized to capture force parameters, as illustrated in Figure 4. Furthermore, the motion analysis software Ortho Trak 6.3.3 was employed in synergy with the “Helen Hayes Marker Set” to precisely position reflective markers, as depicted in Figure 3. The markers were affixed according to the designated nomenclature detailed in Figure 5. In the experiment, participants were asked to pedal at a rate of 50 rpm and 70 rpm, each pedaling for 15 cycles, and the pedaling action was performed a total of 3 times. Using a metronome, the experimenter adjusted the pedaling process with metronome frequencies set at 50 and 70 RPM.

Utilizing the motion analysis system installed on indoor stationary bicycles and elliptical machines, reflective markers were placed to provide phased utilization. The reflective marker is designated as 0° or 360° when it appears at the highest point, 90° at the forward pedal position, 180° at the lowest point during downward pedal motion, and 270° at the rear pedal position. The range from 0° to 180° is defined as Phase 1, and the range from 180° to 360° is defined as Phase 2. For the comprehensive analysis of joint forces and moments during movement, an inverse dynamic analysis method was employed [36]. The net joint torque (NET) was astutely determined by summing the gravitational torque (GRA), the motion-dependent torque (MDT), the contact torque (CT), and the generalized muscle torque (MUS). To accurately obtain the kinematic parameters, a sophisticated coordinate processing technique was applied, encompassing coordinate transformation, data synchronization, and extraction. The motion analysis system captured spatial coordinates and computed spatial matrices in relation to the established laboratory coordinates. Transformed values were obtained using the ATI system, and the ATI signal was meticulously synchronized with the motion analysis system to calculate vital joint kinematic parameters, including the joint angle, angular velocity, and angular acceleration. Abbreviations and detailed explanations of joint motions, forces and torques are in Table 1.

The dynamic analysis research of Zernicke (1986) [36] provided the following derivation:Net joint torque = Gravitational torque + Motion-dependent torque + Contact torque + Generalized muscle torque
Newton: F = m × α
ΣFx = m × ax, ΣFy = m × ay, ΣFy = m × az − m × g
Euler: M = I × α
ΣMx = Ix × ax + Fy × Dy + Fz × Dz
ΣMy = Iy × ay + Fx × Dx + Fz × Dz
ΣMz = Iz × az + Fy × Dy + Fx × Dx

After employing these formulas, numerical values were obtained, as illustrated in Figure 6.

**Figure 6 medicina-60-00498-f006:**
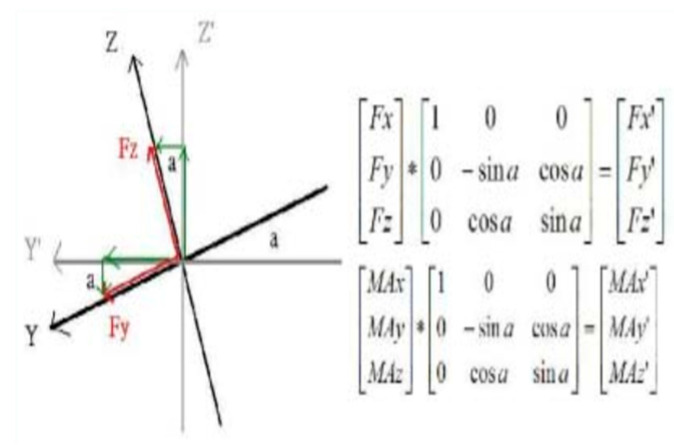
Spatial transformation matrix.

**Table 1 medicina-60-00498-t001:** Abbreviations and detailed explanations for joint movements, forces, and torques.

Abbreviation	Detailed Explanation
L.Ankle_Rot	Left Ankle Joint Rotation
L.Ankle_Abd	Left Ankle Joint Inversion and Eversion
L.Ankle_Flex	Left Ankle Joint Dorsiflexion and Plantarflexion
L.Knee_Rot	Left Knee Joint Internal and External Rotation
L.Knee_Abd	Left Knee Joint Medial and Lateral Rotation
L.Knee_Flex	Left Knee Joint Flexion and Extension
Fx	Medial–Lateral Force of the Pedal in the Horizontal Plane
Fy	Anterior–Posterior Force of the Pedal in the Horizontal Plane
Fz	Vertical Force Acting on the Pedal
Ax	Lateral Force Acting on the Ankle Joint in the Horizontal Plane
Ay	Anterior–Posterior Force Acting on the Ankle Joint in the Horizontal Plane
Az	Vertical Force Acting on the Ankle Joint in the Horizontal Plane
Kx	Lateral Force Acting on the Knee Joint in the Horizontal Plane
Ky	Anterior–Posterior Force Acting on the Knee Joint in the Horizontal Plane
Kz	Vertical Force Acting on the Knee Joint in the Horizontal Plane
MAx	Rotational Torque of the Ankle Joint in the Sagittal Plane
MAy	Medial–Lateral Rotational Torque of the Ankle Joint in the Horizontal Plane
MAz	Rotational Torque of the Ankle Joint in the Horizontal Plane
MKx	Rotational Torque of the Knee Joint in the Sagittal Plane
MKy	Medial–Lateral Rotational Torque of the Knee Joint in the Horizontal Plane
MKz	Rotational Torque of the Knee Joint in the Horizontal Plane

The data were statistically evaluated using SPSS version 23.0. A one-factor repeated-measures analysis of variance (ANOVA) was applied, with the significance level set at α = 0.05, meaning that statistical significance was ascertained at a *p*-value of less than 0.05.

## 3. Results

### 3.1. Significant Differences in Joint Angles between Stationary Bikes and Elliptical Machines

Mean Values of Joint Angles

Upon analyzing the data presented in Table 2, notable distinctions in the mean values of specific joint angles, specifically L_Ank_Rot and L_Ank_Abd, were identified when conducting a comparative analysis between the two types of exercise machines. The observed differences were found to be statistically significant.

### 3.2. Significant Differences in Peak Angle Values for L_Ank_Rot and L_Ank_Abd between Stationary Bikes and Elliptical Machines

#### 3.2.1. Different Pedaling Frequencies: 50 rpm

Paired-sample *t*-tests were employed to assess the differences in L_Ank_Rot and L_Ank_Abd at the same speeds but across distinct exercise machines. The findings illustrated in Table 3 demonstrate statistically significant disparities in peak angle values for both L_Ank_Rot and L_Ank_Abd specifically at 50 rpm (*p* < 0.05).

#### 3.2.2. Different Pedaling Frequencies: 70 rpm

Comparable paired-sample *t*-tests were carried out at a speed of 70 rpm. As outlined in Table 4, the results exhibited noteworthy differences in peak angle values for both L_Ank_Rot and L_Ank_Abd, with statistical significance observed (*p* < 0.05).

### 3.3. The Analysis of the Effect of Movement Speed on Vertical Forces Acting on the Knee Joint in the Horizontal Plane

#### A Comparison of Vertical Forces at Different Speeds on the Knee Joint

An examination of the vertical forces exerted on the knee joint in the horizontal plane revealed significant differences in the parameters solely during Phase 1 across different speeds, as detailed in Table 5. Conversely, no significant differences were observed in any parameters during Phase 2, as depicted in Table 6. The data analysis was conducted using a one-factor repeated-measures ANOVA in SPSS version 23.0, with the significance level set at α = 0.05.

### 3.4. Comparison of Joint Forces and Moments between Elliptical Machine and Stationary Bike

#### Peak Values of Pedal Reaction Force, Ankle Joint Force, and Knee Joint Force in Horizontal and Vertical Planes

This section provides an in-depth analysis and comparison of the peak values of the pedal reaction force, ankle joint force, and knee joint force in both the horizontal and vertical planes across various speeds for both stationary bikes and elliptical machines. Detailed visual representations of these comparisons are presented in Figure 7.

### 3.5. Comparison of Joint Torque between Elliptical Machine and Stationary Bike

#### Peak Joint Torque Angle and Direction

The points at which the joint torque peaks emerged occurred earlier on elliptical machines than on indoor exercise bikes. An analysis and comparison of peak torques at the ankle and knee joints were conducted, as both the elliptical machine and the indoor exercise bike undergo reciprocal motion in the horizontal plane. Figure 8 provides a comprehensive depiction of these comparative analyses.

## 4. Discussion

The elliptical machine is characterized by a fixed pedal pattern, in contrast to the bicycle, which permits variable pedal positions stemming from the trajectory and force applied by the user. These divergences in pedal patterns yielded substantial differences in joint angles between the two types of exercise machines, especially at lower speeds. In particular, the angles of L_Ank_Rot and L_Ank_Abd manifested significant discrepancies (*p* ≤ 0.05). Additionally, the study noted distinct ankle joint movements between the machines: on the elliptical machine, Ank_Rot was associated with internal rotation, while Ank_Abd was linked to inversion. Conversely, on the stationary bike, Ank_Rot was related to external rotation, and Ank_Abd to eversion.

In the examination of joint strength, this study revealed that knee joint strength in the anterior–posterior and vertical directions was more pronounced during pedaling on an elliptical machine than on a stationary bike. However, the peak ankle joint strength in the aforementioned directions was observed earlier on the stationary bike than on the elliptical machine. The timing of peak ankle joint strength in the horizontal direction was similar between the two machines. These insights could guide exercise equipment manufacturers in adjusting linkage positions to reduce peak values and enhance joint strength during exercise. Significant differences were noted in the peak values of the pedal reaction force, ankle joint force, and knee joint force in the vertical direction with the elliptical machine at increased pedaling speeds. The stationary bike did not display significant differences for the same parameters. It was found that when using the elliptical machine at high speeds, maintaining the center of gravity at a central point was vital to prevent excessive rocking, which could lead to augmented forces acting on the pedal, ankle joint, and knee joint in the vertical direction. This finding is congruent with the results reported by Gottschall et al. (2005) [37], emphasizing that an increase in the inclination angle of the horizontal plane generates greater ground reaction forces. These observations were made during the second phase of the elliptical exercise in this study.

Furthermore, the trajectory of the fitness equipment had a pronounced impact on the detected differences in vertical force between pedaling on the elliptical machine and cycling. This discrepancy can be ascribed to the elliptical machine’s unique trajectory, which introduced a tilt. Moreover, the joint angles shed light on the extent of the pedal reaction force, ankle joint force, and knee joint force in the horizontal plane. Specifically, pedaling on the elliptical machine led to the internal rotation and inversion of the ankle joint, while the stationary bike induced external rotation and eversion. These disparate ankle joint motions may have contributed to the peak values of the pedal reaction force, ankle joint force, and knee joint force in the horizontal plane. Importantly, the fixed pedal of the elliptical machine used in this experiment constrained the ankle joint angle, necessitating an increased knee joint load to counteract the anterior–posterior torque in the horizontal plane.

The peak joint torque during cycling was observed to occur earlier in the pedal stroke, specifically between 70° and 110°, compared to the elliptical machine, where it manifested between 135° and 180°. This distinction is attributable to the seated position in cycling versus the standing position on the elliptical machine, resulting in a shift in the center of gravity with the elliptical machine and thus a delayed onset of joint torque. Furthermore, the peak ankle and knee joint torques in the horizontal plane during the use of the elliptical machine were observed later than the peak ankle torque, specifically between 160° and 180°. This timing coincides with the later onset of the pedal reaction force in the horizontal plane. Previous research by Jian disclosed that the peak ankle torque in the horizontal plane is the lowest among all directions in the joint torque curve of the elliptical machine [38], corroborating the findings of the present study. Significant variations in torque values at the ankle joint were found due to the differences in joint angles between the two exercise machines. Additionally, the peak joint torque for all joints was elevated during high-speed pedaling compared to low-speed pedaling for both machines. Therefore, to avert joint injuries, the ability to tolerate the pace of the exercise machine must be considered when utilizing these devices.

This study leveraged a biomechanical approach to analyze the net joint torque, encompassing several factors but omitting joint friction torque due to the non-invasive measurement system utilized. It is vital to recognize that the joint friction torque was not included in the analysis when interpreting and applying the study’s findings to similar exercise equipment. Furthermore, variations in design and specifications among different products should be regarded when extrapolating the results of this study.

It is worth noting that the position of the pedal changes throughout the pedal stroke in cycling, leading to variations in ankle joint angles and a heightened risk of injury. Conversely, on elliptical machines, the peak values of pedal reaction force, ankle joint force, and knee joint force occur later in the pedal stroke compared to stationary bikes. As the pedaling speed on elliptical machines increases, there is a corresponding rise in the peak values of these forces in the vertical direction. This emphasizes the importance of maintaining a centered gravity to avoid excessive rocking and diminish the effects of increased forces on the pedal, ankle joint, and knee joint. The fixed-pedal design of the elliptical machine limits ankle joint angles, causing internal rotation and requiring an augmented knee joint load to offset anterior–posterior force in the horizontal plane. Both types of exercise machines show higher peak joint torque values at high speeds relative to low speeds, with stationary bikes exhibiting earlier peak values than elliptical machines. To prevent sports injuries, selecting an appropriate speed for either type of fitness equipment is crucial. Acknowledging variations in product design and specifications is essential when extrapolating the findings of this study to similar exercise equipment. Additionally, since joint friction torque is not considered due to the lack of invasive measurement systems, caution is advised when interpreting and applying the results of this study.

By investigating the use of stationary bikes and elliptical machines to examine knee joint movement and joint load during exercise, individuals can fine-tune their workouts to enhance performance and prevent injuries. The data on peak torque at various pedaling speeds and machine types enable users to optimize their exercise strategies. Furthermore, understanding how equipment selection influences joint kinetics empowers users to make informed choices, potentially reducing the risk of knee injuries. In summary, these research findings offer evidence-based guidance for designing effective and safe exercise plans tailored to individual needs and preferences. Manufacturers can translate these into clinical recommendations by focusing on equipment design, user guidelines, safety measures, and user training. Specifically, they should design elliptical machines with adjustable settings to accommodate a wider range of users and mitigate higher peak torques. Clear guidelines should be provided for appropriate equipment usage based on individual characteristics, with consideration given to potential risks to the knee joint. Implementing safety measures, such as warning labels and instructional materials, can help educate users about potential risks and proper usage techniques. Additionally, offering user training programs ensures that fitness professionals and physical therapists can effectively guide individuals in using the equipment safely and optimizing exercise routines. These efforts aim to enhance user safety and optimize the effectiveness of stationary bikes and elliptical machines for diverse user populations.

## 5. Conclusions

This study offers insight into the divergent joint torque patterns between the elliptical machine and the stationary bike. The elliptical machine demonstrated higher peak joint torque values in the horizontal plane, manifesting later in the pedal stroke as opposed to the stationary bike. Additionally, significant differences were observed in ankle joint angles, with the elliptical machine showing inward rotation and the stationary bike presenting outward rotation. These results emphasize the need to consider joint angles and the speed of exercise to mitigate the potential risk of joint injuries when employing these types of fitness equipment. Further studies are warranted to examine the effect of joint friction torque on joint torque during exercise and to explore design enhancements that could align ankle joint angles and reduce potential joint stress.

The ankle joint angle during elliptical machine use may experience a phenomenon of inversion, whereas substantial variations in the ankle joint angle occur on the stationary bike. To address this concern, future elliptical machine designs could be modified to align with the pedaling trajectory and reduce joint loading. These design considerations would necessitate further research to optimize aspects that can promote the alignment of ankle joint angles and alleviate undue joint strain during exercises on the elliptical machine. Despite the original intention of reducing impact, the experimental findings uncovered considerable torque generation in both the horizontal and vertical planes during elliptical machine use. By moving away from fixed pedals and adopting a biomechanically oriented design that corresponds with human movement patterns, there is potential to further lessen the impact and joint loading. Such design innovation could provide a competitive edge in the fitness equipment market. In future experiments, incorporating force sensors on both the left and right pedals may facilitate the simultaneous measurement of torque phenomena from both feet. This approach could generate more accurate data, thereby enhancing our comprehension of the pedaling mechanics during exercise and contributing to the development of more ergonomically sound exercise equipment.

## Figures and Tables

**Figure 1 medicina-60-00498-f001:**
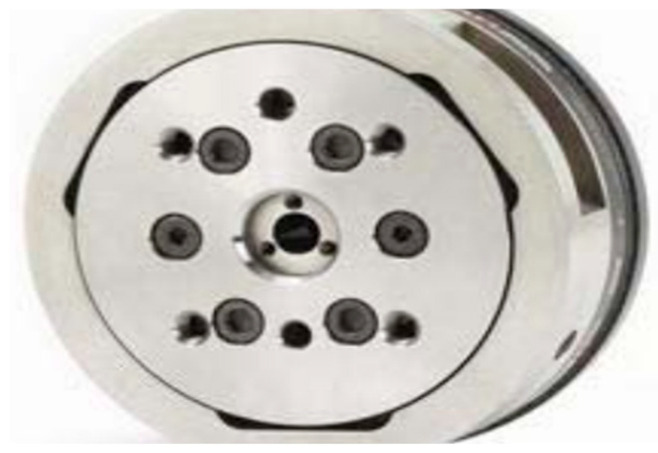
Six-Axis/Torque Sensor System A3, ATI, USA.

**Figure 2 medicina-60-00498-f002:**
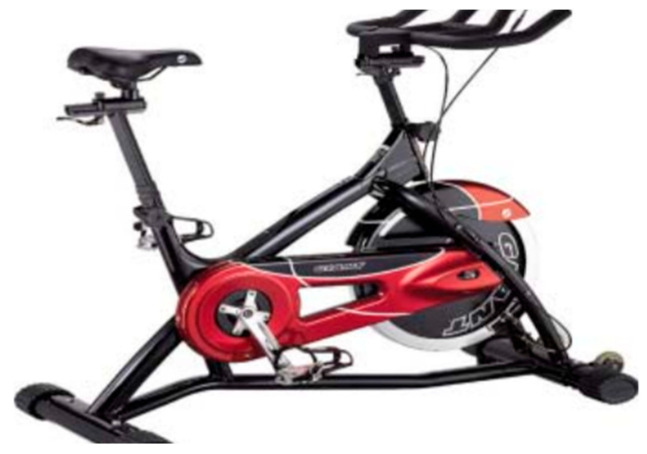
Stationary bicycle/Giant tempo indoor exercise bicycle, Taiwan.

**Figure 3 medicina-60-00498-f003:**
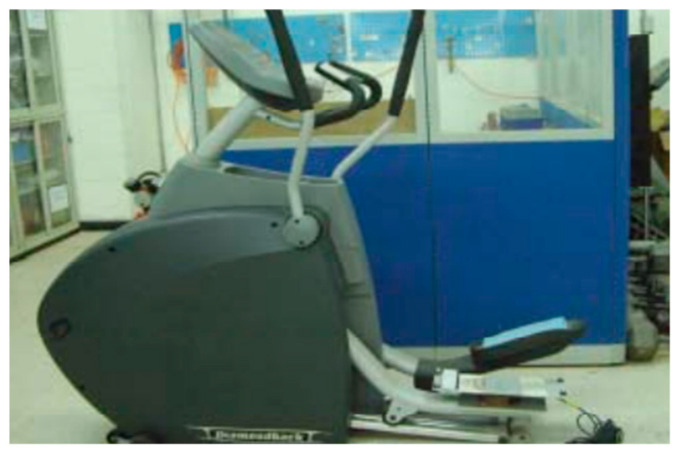
Elliptical machine/Diamondback 460EF, USA.

**Figure 4 medicina-60-00498-f004:**
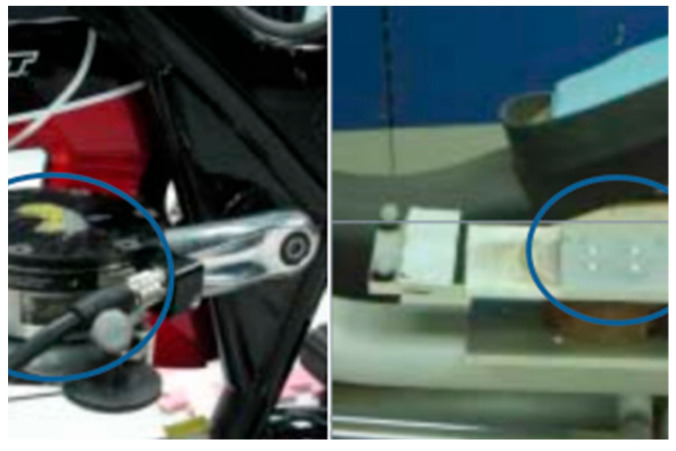
The installation position of the ATI on a stationary bicycle and elliptical machine, as well as the adhesive location of the reflective ball. Note: The portion circled in blue is the ATI.

**Figure 5 medicina-60-00498-f005:**
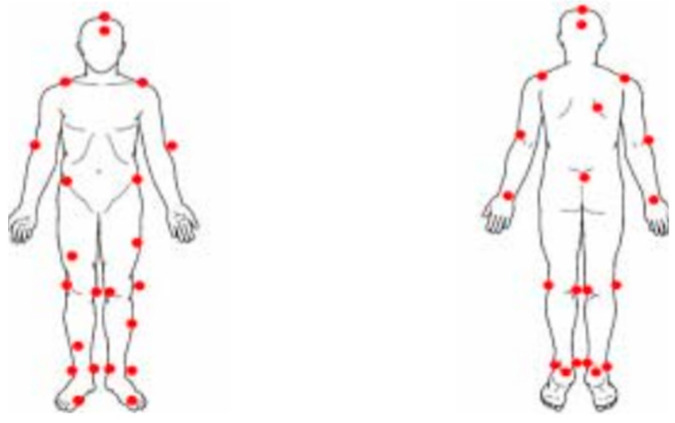
Pasting positions of reflective balls.

**Figure 7 medicina-60-00498-f007:**
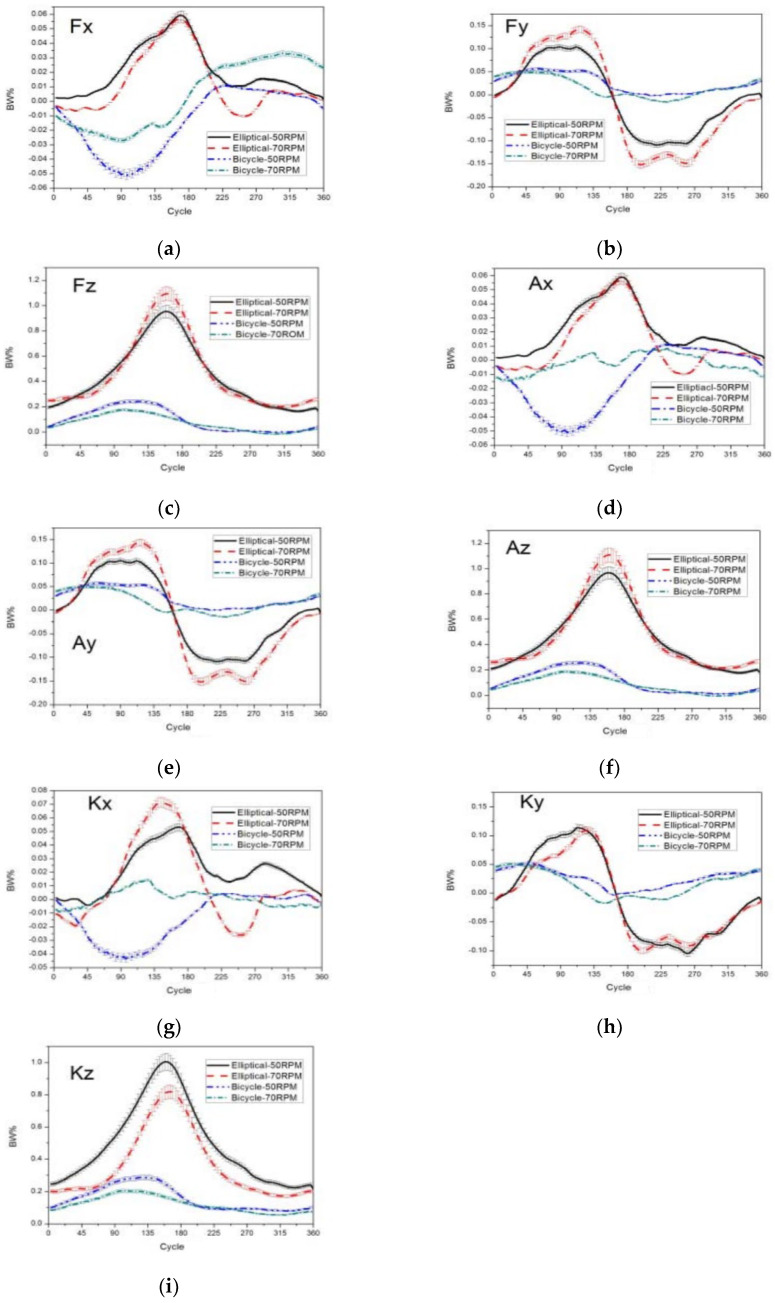
Graphs of the stationary bike and the elliptical machine at different speeds. The figure is divided into the following subparts: (**a**) the pedal reaction force direction differed between the stationary bike and the elliptical machine due to ankle inversion and eversion; (**b**) in Phase 2, the peak pedal reaction force in the horizontal plane was directed forward but became negative after 180 degrees; (**c**) the peak pedal reaction force in the vertical direction occurred later on the elliptical machine than on the stationary bike; (**d**) stable peak ankle joint values in the horizontal plane during high-speed curves while riding the stationary bike were attributed to the need for joint stability at high speeds to keep up with the pace; (**e**) peak ankle joint values in the horizontal plane were higher for the elliptical machine than for the stationary bike in both phases; (**f**) the curve of the ankle joint in the front–back direction in the horizontal plane was similar to that of the pedal reaction force; (**g**) at high speed, the peak value of the knee joint in the left–right direction in the horizontal plane was significantly increased for the elliptical machine; (**h**) there was no significant difference in the curve of the knee joint in the front–back direction in the horizontal plane at different speeds for both types of equipment; (**i**) the peak value of the knee joint in the vertical direction in the horizontal plane was about 1 BW for the elliptical machine, which was significantly greater than that for the stationary bike.

**Figure 8 medicina-60-00498-f008:**
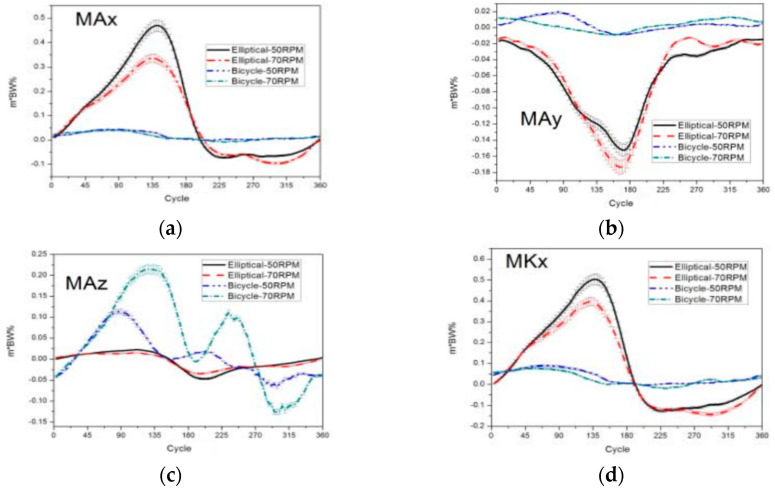
Graphs of the stationary bike and the elliptical machine at different speeds. The figure is divided into the following subparts: (**a**,**d**) The stationary bike exhibits an earlier occurrence of peak joint torque compared to the elliptical machine. The peak joint torque of the ankle and knee joints in the sagittal plane during flexion and extension is greater on the elliptical machine than on the stationary bike. (**b**,**e**) Moreover, the peak joint torque of the ankle and knee joints in the horizontal plane during internal and external rotation is greater on the elliptical machine than on the stationary bike. (**c**,**f**) The peak joint torque of ankle and knee joints during internal and external rotation in the horizontal plane occurs in both Phase 1 and Phase 2 while riding the stationary bike.

**Table 2 medicina-60-00498-t002:** Lower limb joint angles for the left knee and ankle joint at different pedaling frequencies (50 rpm and 70 rpm) using various fitness equipment (unit: degree).

	50 rpm		70 rpm		*p*
	Elliptical Machine	Stationary Bike	Elliptical Machine	Stationary Bike	
Ankle_Rot	−14.51 (11.25)	100.39 (76.47)	−20.98 (17.55)	120.06 (84.19)	0.004 *
Ankle_Abd	−23.60 (21.17)	59.30 (16.74)	−25.66 (16.65)	51.53 (37.28)	0.001 *
Ankle_Flex	69.14 (8.69)	44.43 (42.37)	62.25 (16.28)	38.48 (37.89)	0.065
Knee_Rot	12.67 (25.84)	38.81 (63.81)	22.79 (16.36)	74.32 (57.17)	0.102
Knee_Abd	40.02 (9.39)	53.91 (69.71)	26.70 (22.04)	28.98 (35.38)	0.834
Knee_Flex	156.74 (5.24)	66.16 (94.02)	149.38 (12.04)	99.58 (78.99)	0.689

The asterisk (*) in the statistical table indicates statistical significance.

**Table 3 medicina-60-00498-t003:** Lower limb joint angles for the left ankle joint at a pedaling frequency of 50 rpm using various fitness equipment (unit: degree).

	50 rpm		*p*
	Elliptical Machine	Stationary Bike	
Ankle_Rot	−14.51 (11.25)	100.39 (76.47)	0.012 *
Ankle_Aad	−23.60 (21.17)	59.30 (16.74)	0.002 *

The asterisk (*) in the statistical table indicates statistical significance.

**Table 4 medicina-60-00498-t004:** Lower limb joint angles for the left ankle joint at a pedaling frequency of 70 rpm using various fitness equipment (unit: degree).

	70 rpm		*p*
	Elliptical Machine	Stationary Bike	
Ankle_Rot	−20.98 (17.55)	120.06 (84.19)	0.005 *
Ankle_Aad	−25.66 (16.65)	51.53 (37.28)	0.003 *

The asterisk (*) in the statistical table indicates statistical significance.

**Table 5 medicina-60-00498-t005:** Lower limb reverse force and joint strength for the left knee and ankle joint at different pedaling frequencies (50 rpm and 70 rpm) using various fitness equipment (Phase 1, unit: BW%).

	50 rpm		70 rpm		*p*
	Elliptical Machine	Stationary Bike	Elliptical Machine	Stationary Bike	
Fx	0.05 (0.04)	−0.001 (0.01)	0.05 (0.06)	0.01 (0.03)	0.700
Fy	0.13 (0.05)	0.07 (0.04)	0.07 (0.04)	0.07 (0.04)	0.133
Fz	0.89 (0.27)	0.24 (0.09)	0.24 (0.09)	0.24 (0.09)	0.121
Ax	0.05 (0.04)	−0.001 (0.01)	0.05 (0.06)	0.02 (0.06)	0.483
Ay	0.13 (0.05)	0.07 (0.04)	0.15 (0.09)	0.09 (0.05)	0.129
Az	0.91 (0.27)	0.26 (0.09)	1.08 (0.31)	0.25 (0.11)	0.126
Kx	0.05 (0.04)	0.00 (0.01)	0.05 (0.06)	0.02 (0.07)	0.280
Ky	0.13 (0.06)	0.06 (0.04)	0.17 (0.09)	0.08 (0.04)	0.059
Kz	0.94 (0.28)	0.29 (0.09)	1.20 (0.26)	0.27 (0.14)	0.001 *

The asterisk (*) in the statistical table indicates statistical significance.

**Table 6 medicina-60-00498-t006:** Lower limb reverse force and joint strength for the left knee and ankle joint at different pedaling frequencies (50 rpm and 70 rpm) using various fitness equipment (Phase 2, unit: BW%).

	50 rpm		70 rpm		*p*
	Elliptical Machine	Stationary Bike	Elliptical Machine	Stationary Bike	
Fx	0.05 (0.04)	0.01 (0.01)	0.04 (0.05)	0.05 (0.07)	0.158
Fy	0.01 (0.05)	0.03 (0.02)	−0.00 (0.04)	0.04 (0.03)	0.660
Fz	0.80 (0.15)	0.06 (0.11)	0.81 (0.27)	0.09 (0.07)	0.817
Ax	0.05 (0.04)	0.01 (0.01)	0.04 (0.05)	0.03 (0.03)	0.402
Ay	0.01 (0.05)	0.02 (0.02)	−0.00 (0.04)	0.03 (0.02)	0.642
Az	0.82 (0.15)	0.08 (0.11)	0.83 (0.28)	0.10 (0.07)	0.816
Kx	0.05 (0.04)	0.01 (0.01)	0.05 (0.05)	0.03 (0.04)	0.177
Ky	0.02 (0.05)	0.04 (0.02)	−0.01 (0.06)	0.04 (0.03)	0.505
Kz	0.86 (0.15)	0.13 (0.11)	0.88 (0.21)	0.14 (0.09)	0.962

## Data Availability

The data used to support the findings of this study are included within the article.

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
