# Peer review of "Effects of Stationary Bikes and Elliptical Machines on Knee Joint Kinematics during Exercise"

_medicina, 2024, doi:10.3390/medicina60030498_

Round 1

Reviewer 1 Report

Comments and Suggestions for Authors

The described research indicates the differences that occur in the angular movements and loading of the knee and ankle joints when using a stationary bicycle compared to an elliptical trainer.

Measurement tools for performing this task were provided, but the type and brand of equipment on which the research was conducted was not specified. The various market products may have different parameters and therefore the data of the equipment used should be given.

In addition, the conclusions, while important for manufacturers, were not translated into clinical recommendations. For whom the different types of equipment should be intended (body characteristics, e.g., postural defects, medical conditions, etc.), and which people (patients) should not use them.

The type of equipment described is popular both in fitness centers and at home, and can be used inappropriately, harming users.

Author Response

Reviewer 1

Comments and Suggestions for Authors

The described research indicates the differences that occur in the angular movements and loading of the knee and ankle joints when using a stationary bicycle compared to an elliptical trainer.

Measurement tools for performing this task were provided, but the type and brand of equipment on which the research was conducted was not specified. The various market products may have different parameters and therefore the data of the equipment used should be given.

Response: Thank you very much for the suggestions you provided to me. I have supplemented the relevant data regarding the types and brands of research equipment on line 120 to 122 of the article. Your suggestions have made my article more rigorous, and I appreciate your input.

In addition, the conclusions, while important for manufacturers, were not translated into clinical recommendations. For whom the different types of equipment should be intended (body characteristics, e.g., postural defects, medical conditions, etc.), and which people (patients) should not use them.

Response: In the discussion section, I have incorporated your suggestions from line 351 to line 370, which have enhanced the logical coherence and overall flow of the article. I am truly grateful for your valuable advice.

The type of equipment described is popular both in fitness centers and at home, and can be used inappropriately, harming users.

Response: I have thoroughly discussed this issue from line 351 to line 370, hoping that people can achieve positive exercise outcomes and avoid injuries when using exercise equipment. Thank you for your valuable suggestions.

Reviewer 2 Report

Comments and Suggestions for Authors

I congratulate the authors for their study. The research is original and interesting to read. The flow of the theoretical, as well as other significant parts of the paper, creates a sense of ingenuity, which engages the reader. The language is clear and academic, without errors. The materials and methods section is detailed, with subsections covering sampling, data collection, and participants (data analysis and ethical considerations). All significant aspects are delineated and well-expounded upon. The organization of the article is relevant. However, I have also stated my opinions and suggestions below in order to increase the quality of the study.

The stationary bikes and elliptical machines are devices with different characteristics, for example, in bicycles, performance is performed in a sitting position, while in elliptical machines, performance is performed in a standing position. This situation causes differences in the load on the joints and muscles and in the patterns of muscle contraction. The authors should better explain why they are comparing these two systems. In addition, the practical contribution of the findings obtained in the study to those who will use these systems should be explained.

Had the participants performed on these devices before? Was familiarization done before the study? If the study participants are not accustomed to these devices, this may have affected your measurements.

Additions can be made to the discussion section, considering the suggestions I mentioned above.
